# Harnessing the Dual Antimicrobial Mechanism of Action with Fe(8-Hydroxyquinoline)_3_ to Develop a Topical Ointment for Mupirocin-Resistant MRSA Infections

**DOI:** 10.3390/antibiotics12050886

**Published:** 2023-05-10

**Authors:** Nalin Abeydeera, Bogdan M. Benin, Khalil Mudarmah, Bishnu D. Pant, Guanyu Chen, Woo Shik Shin, Min-Ho Kim, Songping D. Huang

**Affiliations:** 1Department of Chemistry and Biochemistry, Kent State University, Kent, OH 44240, USA; nkekiriw@kent.edu (N.A.); kmudarma@kent.edu (K.M.); bpant@kent.edu (B.D.P.); gchen10@kent.edu (G.C.); 2Department of Pharmaceutical Sciences, College of Pharmacy, Northeast Ohio Medical University, Rootstown, OH 44272, USA; bbenin@neomed.edu; 3Department of Chemistry, Jazan University, Jazan 45142, Saudi Arabia; 4Department of Biological Sciences, Kent State University, Kent, OH 44240, USA; mkim15@kent.edu

**Keywords:** Fe-based antimicrobials, methicillin-resistant *Staphylococcus aureus*, metal chelation, Fenton reaction, bacterial iron metabolism

## Abstract

8-Hydroxyquinoline (8-hq) exhibits potent antimicrobial activity against *Staphylococcus aureus* (SA) bacteria with MIC = 16.0–32.0 µM owing to its ability to chelate metal ions such as Mn^2+^, Zn^2+,^ and Cu^2+^ to disrupt metal homeostasis in bacterial cells. We demonstrate that Fe(8-hq)_3_, the 1:3 complex formed between Fe(III) and 8-hq, can readily transport Fe(III) across the bacterial cell membrane and deliver iron into the bacterial cell, thus, harnessing a dual antimicrobial mechanism of action that combines the bactericidal activity of iron with the metal chelating effect of 8-hq to kill bacteria. As a result, the antimicrobial potency of Fe(8-hq)_3_ is significantly enhanced in comparison with 8-hq. Resistance development by SA toward Fe(8-hq)_3_ is considerably delayed as compared with ciprofloxacin and 8-hq. Fe(8-hq)_3_ can also overcome the 8-hq and mupirocin resistance developed in the SA mutant and MRSA mutant bacteria, respectively. Fe(8-hq)_3_ can stimulate M1-like macrophage polarization of RAW 264.7 cells to kill the SA internalized in such macrophages. Fe(8-hq)_3_ exhibits a synergistic effect with both ciprofloxacin and imipenem, showing potential for combination therapies with topical and systemic antibiotics for more serious MRSA infections. The in vivo antimicrobial efficacy of a 2% Fe(8-hq)_3_ topical ointment is confirmed by the use of a murine model with skin wound infection by bioluminescent SA with a reduction of the bacterial burden by 99 ± 0.5%, indicating that this non-antibiotic iron complex has therapeutic potential for skin and soft tissue infections (SSTIs).

## 1. Introduction

As an opportunistic pathogen, *Staphylococcus aureus* (SA) is the most common cause of skin and soft tissue infections (SSTIs) as ca. 30% of people are colonized by SA on the skin and in the nares [1,2]. SSTIs were once readily treatable with over-the-counter topical ointments consisting of either one or a combination of three common topical antibiotics including bacitracin, neomycin and polymyxin B. However, such days are now gone due to the emergence of methicillin-resistant SA (i.e., MRSA) [3,4]. Currently, there are only two topical antibiotics that are still effective against MRSA: mupirocin (Bactroban^®^) and fusidic acid or fusidate. In the era of rising antimicrobial resistance (AMR), an increasing number of MRSA strains are found to be resistant to mupirocin due in large part to its widespread and routine use in the community and hospitals for nasal SA decolonization [5]. The situation of fusidate resistance found in MRSA strains is even worse [6,7,8]. As the development of resistance to fusidate involves a single point mutation, the generic barriers to mutation are low, diminishing the efficacy of fusidate as a topical monotherapy for treating SSTIs. As a result, fusidate has never been approved for use as a topical antibiotic in the U.S. but remains common in Europe. When faced with the therapeutic uncertainty of using these two topical antibiotics to treat SSTIs, physicians and dermatologists often prescribe a broad-spectrum oral or injectable fluoroquinolone antibiotic in combination with one of the two topical ointments to improve treatment outcomes. This practice inadvertently exerts selective pressure for the ciprofloxacin-resistant MRSA phenotypes to prevail [9,10]. It should be noted that both of these topical antibiotics are prescription-only medications, which excludes some patients from seeking treatment for their seemingly harmless SSTIs. However, if not treated in a timely manner, some of these patients may develop more serious and life-threatening systemic infections as SSTIs are often the bacterial point of entry into the human systems.

This study aims to develop novel topical antimicrobial agents for treating mupirocin-resistant and fusidate-resistant SSTIs by MRSA, particularly by the mupirocin-resistant and fusidate-resistant MRSA. We focus on the design and synthesis of metal-complexes that can target cellular and molecular components that are different from those targeted by conventional antibiotics [11,12,13,14,15]. In this publication, we demonstrate that Fe(8-hq)_3_ (8-hq = 8-hydroxyquinoline): (1) exhibits potent antimicrobial activity against four different strains of SA bacteria with the ability to eradicate such bacteria at a relatively low concentration (e.g., 2 μM for 1-log reduction); (2) inhibits MRSA biofilm formation; (3) stimulates M1-like macrophage polarization of RAW 264.7 cells to kill SA bacteria internalized in the macrophages; (4) shows considerably delayed resistance development and the ability to overcome the drug resistance exhibited by 8-hq-, ciprofloxacin-, mupirocin- and fusidate-resistant mutant SA bacteria; (5) has a synergistic effect with both ciprofloxacin and imipenem, signifying potential for use in combination therapies with these antibiotics; and (6) shows in vivo efficacy in a murine excisional wound skin infection model with a 2% Fe(8-hq)_3_ topical ointment, demonstrating the therapeutic potential of this iron complex for skin and soft tissue infections (SSTIs). Overall, Fe(8-hq)_3_ has all the desirable characteristics for use as a new topical antimicrobial agent to treat mupirocin-resistant MRSA SSTIs and would be a promising over-the-counter (OTC) medication in place of topical mupirocin or fusidate given its low likelihood to develop cross-resistance from or to conventional antibiotics.

## 2. Results and Discussion

### 2.1. Synthesis and Characterization of Fe(8-hq)3

The reaction between FeCl_3_ and 8-hydroxyquinoline, in a molar ratio of 1:3 in ethanol at room temperature, afforded a greenish-black precipitate after 3 h of vigorous stirring (Appendix A). We isolated the product by filtration and washing with ethanol three times. Due to the paramagnetic nature of this complex, ^1^H NMR studies were not carried out. Instead, characterization by X-ray powder diffraction (PXRD; Figure 1a), UV-Vis (Appendix A), FT-IR (Appendix A), elemental analysis (Appendix A) and LC-MS (Appendix A) clearly confirmed that the identify of this product was Fe(8-hq)3 with purity of ≥98%. As shown in Figure 1b,c, the crystal structure analysis of this complex revealed that Fe^3+^ is fully encapsulated in the octahedral cavity created by three O- and three N-donor atoms from three lipophilic aromatic 8-hq molecules, effectively concealing the ionic nature of the highly charged Fe(III) center. Additionally, the *D*3 molecular symmetry annuls any potential overall molecular polarity in this octahedral Fe(III)-complex. As inferred by the molecular symmetry, Fe(8-hq)_3_ should exist as a racemic mixture of two chiral enantiomers. However, we made no attempt in this work to obtain chiral enantiomers by chiral separation for all the bioactivity studies.

### 2.2. Evaluation of In Vitro Antibacterial Activity against SA

The minimum inhibitory concentration (MIC) was determined in four different strains of SA bacteria using the broth microdilution technique. The bacteria under investigation included a methicillin-susceptible strain of SA (MSSA; ATCC 6538), two methicillin-resistant strains of SA (MRSA^α^; ATCC BAA-44 and MRSA^β^; USA 300, ATCC BAA-1717) and a vancomycin-intermediate strain of SA (VISA; ATCC 700699). The MIC values of Fe(8-hq)_3_ were measured concurrently with those of 8-hq for comparison in these strains of bacteria. The results are summarized in Table 1. It is well-known that 8-hq has rather widespread antifungal [16], antibacterial [17], antiviral [17] and anticancer activity [18], owing to its ability to chelate metal ions such as Mn^2+^, Zn^2+^ and Cu^2+^, which can disrupt intracellular metal homeostasis [19]. However, this single antimicrobial mechanism of action can readily result in the development of 8-hq resistance in bacteria as revealed by the data from the current study (*vide infra*). To ascertain whether the complexation of 8-hq with Fe(III) has a clear enhancement of antimicrobial activity, we studied the dose-dependent response of the four strains toward Fe(8-hq)_3_ using the colony-forming unit (CFU) enumeration technique. For comparison, the dose-dependent response of the corresponding bacterial cells treated with 8-hq alone was determined side-by-side in separate agar plates. 

The results showed that, in MSSA, an approximately 1-log reduction of bacterial colonies could be achieved by Fe(8-hq)_3_ at a concentration of 2.0 μM. In contrast, 3-log and 5-log reductions were reached by 4.0 μM and 8.0 μM, respectively (Figure 2a and Appendix A). The comparison of antimicrobial activity between Fe(8-hq)_3_ and 8-hq uncovered a remarkable difference, i.e., Fe(8-hq)_3_ was found to be approximately four orders of magnitude more potent than 8-hq in their ability to eradicate MSSA bacteria. Similar trends were observed in the data for the other three strains of SA bacteria. For example, in MRSA^α^, an approximately 2.5-log reduction of bacterial colonies could be achieved by Fe(8-hq)_3_ at a concentration of 4.0 μM, while a 4-log and 5-log reduction were reached at the concentration of 8.0 μM and 16.0 μM, respectively (Figure 2b and Appendix A). In MRSA^β^, an approximately 2.5-log reduction of bacterial colonies could be achieved by Fe(8-hq)_3_ at the concentration of 4.0 μM, while a 5-log and 6-log reduction were reached at the concentration of 8.0 μM and 16.0 μM, respectively (Figure 2c and Appendix A). VISA appeared to be less susceptible to Fe(8-hq)_3_ than MSSA and the two strains of MRSA, with an approximately 1.5-log reduction of bacterial colonies achieved by Fe(8-hq)_3_ at a concentration of 4.0 μM, while a 4-log and 5-log reduction were reached at the concentration of 8.0 μM and 16.0 μM, respectively (Figure 2d and Appendix A). These results, therefore, demonstrate that complexation of Fe(III) with 8-hq substantially enhances antimicrobial activity. Overall, our results confirm that the potent antimicrobial activity of this complex may be attributable to its dual mechanism of action, i.e., a “push-and-pull” effect of simultaneously transporting Fe(III) across the bacterial cell membrane (“push”) and chelating intracellular bio-essential metal ions (“pull”). Such effect was further manifested in the different profiles of resistance development between Fe(8-hq)_3_ and 8-hq (*vide infra*). 

### 2.3. Time–Kill Assay

Time–kill curves of Fe(8-hq)_3_ were obtained for MRSA^α^. We found that Fe(8-hq)_3_ was bacteriostatic when the dose was below 1 × MIC, while it was bactericidal when the dose was above 2 × MIC towards MRSA^α^. In the range of the bactericidal concentrations, Fe(8-hq)_3_ exhibited rather slow killing so that a significant CFU reduction only occurred after 6 h when the bacteria were treated with the complex (Figure 3a).

### 2.4. Cellular Uptake Studies

We performed the study of cell membrane penetration by Fe(8-hq)_3_ to reveal its potential mode of action. The cellular uptake of this complex in MRSA^α^ was analyzed using the atomic absorption spectrometry (AAS) for Fe concentrations in the cell lysates. First, bacterial cells were treated with Fe(8-hq)_3_ at two different subinhibitory concentrations (the treatment group) or with FeCl_3_ at the corresponding concentrations (the reference group), while the control group contained untreated bacterial cells. To ensure that there were enough viable cells to uptake Fe(8-hq)_3_, cells were incubated with the complex for only 6 h. The results were normalized with the population of live bacterial cells and revealed that there was a dose-dependent increase of intracellular Fe contents by about three times at the concentration of 8 μM. As expected from the ionic nature of Fe(III), the reference group, i.e., the group containing the bacteria treated with FeCl_3_ exhibited no increase in the intracellular Fe contents (Figure 3b). 

### 2.5. Inhibition of MRSA^α^ Biofilm Formation

The enhanced growth-inhibitory effect of Fe(8-hq)_3_ in the planktonic MRSA^α^ bacteria prompted us to investigate its potential anti-biofilm activity. It is known that biofilm-derived bacteria are more resistant to the attack of conventional antibiotics in comparison with their planktonic counterparts [20,21,22,23]. To test the potential activity of Fe(8-hq)_3_ against biofilm-derived MRSA^α^, we used the assay previously reported in the literature to measure such inhibitory activity [23]. The results of CFU enumeration studies showed that Fe(8-hq)_3_ inhibited the growth of MRSA^α^ bacteria derived from biofilms with the onset of 0.5 μM (0.125 × MIC) to afford 43% inhibition, while >99% inhibition was achieved at the concentration of 4.0 μM (1 × MIC) (Appendix A). It should be noted that 8-hq itself exhibits considerable activity against the biofilm-derived MRSA^α^ bacteria, but Fe(8-hq)_3_ can eradicate these bacteria at the concentration of 4.0 μM.

### 2.6. Measurements of Intracellular Generation of ROS

We hypothesized that the enhanced antibacterial activity of Fe(8-hq)_3_ originates mostly from the ability of 8-hq to deliver Fe(III) across the cell membrane and release Fe(II) to the intracellular free iron store when the metal centered is reduced, which in turn triggers a Fenton reaction to produce intracellular ROS. Hence, the intracellular levels of ROS in MRSA^α^ bacteria treated with varying concentrations of this complex were measured using the cellular ROS assay based on DCFH-DA (2′,7′-dichlorofluorescein diacetate). The results, after normalization with the live cell population, revealed a dose-dependent generation of ROS associated with a concomitant decrease in live cell population (Figure 4a). This indicates that the intracellular ROS generation is primarily responsible for the bacterial cell death. To further confirm the role of ROS generation in cell death, we performed a rescue experiment using thiourea (TU)—an effective ROS scavenger that can protect cells from the ROS attack by reducing the intracellular accumulation of ROS. As expected, in the presence of TU, the bacterial cell viability of MRSA^α^ was almost fully restored, indicating that such bacterial cells were protected from the ROS attack (Figure 4b). Additionally, the ability of 2,2′-bipyridine (bipy) to inhibit the intracellular generation of ROS was experimentally tested. The latter can effectively quench the Fenton catalytic activity of free Fe(II) through chelation. Once again, in the presence of bipy, the bacterial cell viability of MRSA^α^ was fully restored, indicating that the intracellular ROS generation could be completely blocked and that the free Fe(II) ions released by Fe(8-hq)_3_ were the most likely origin of the ROS attack (Figure 4c).

### 2.7. Imaging Studies of Bacterial Cells by SEM

MRSA^α^ cells were first treated with 8-hq or Fe(8-hq)_3_ before the SEM imaging studies were performed, while the control group contained untreated cells. The most salient feature of the SEM images of the cells treated with Fe(8-hq)_3_ was the presence of a protein matrix covering the cell surfaces as the result of the leaked cytosol, while an obvious change of cellular morphology was visible in the treatment group, indicating that the integrity of cell membrane was compromised. In sharp contrast, the SEM images of the cells treated with 8-hq showed no sign of such damage, indicating that their cell membranes remained intact. The observed cell membrane damage in the treatment group by Fe(8-hq)_3_ is in full agreement with the known cell membrane-lytic properties of intracellular ROS generation (Figure 5a).

### 2.8. The Damaging Effect of Fe(8-hq)_3_ on Cell Membrane of MRSA^α^

The potential damaging effect of Fe(8-hq)_3_ on bacterial cell membrane was investigated by quantitatively measuring the uptake of propidium iodide (PI) in MRSA^α^ bacteria treated with Fe(8-hq)_3_ at three different concentrations for 60 min in comparison with the cells treated with 8-hq at the corresponding concentrations. The PI dye can fluoresce upon binding to nucleic acids but is membrane impermeable and it will therefore remain dark unless the membrane integrity has been compromised. The treatment of MRSA^α^ with Fe(8-hq)_3_ significantly increased the uptake of the PI dye (at 16 μM and 32 μM) as revealed by the increased PI fluorescence signals, confirming the membrane-lytic activity of Fe(8-hq)_3_, while the treatment with 8-hq at the corresponding concentrations exhibited negligible effect on the uptake of PI (Figure 5b,c). The results from these studies and those from SEM imaging studies are consistent with one another.

### 2.9. Bactericidal Activity of RAW 264.7 Macrophages Promoted by Fe(8-hq)_3_ against Internalized SA Bacteria

M1-like macrophages are proinflammatory and microbiocidal owing to their response to scores of different signals from environmental cues to produce ROS and nitric oxide (NO) [24]. Macrophage polarization toward this phenotype is regulated by complex regulatory pathways and constitutes a critical part of innate immune defense against bacterial pathogens [25]. Recently, we have shown that internalized iron oxide nanoparticles (IONPs) in murine macrophage cells (RAW 264.7) can trigger the killing of internalized bacteria as the M1-like macrophage polarization can be stimulated via the Fenton catalytic reaction of iron [26]. To investigate whether the Fenton catalytic reaction inside RAW 264.7 cells could be triggered by Fe(8-hq)_3_, we evaluated the effect of the intracellular redox signaling on the promotion of bacterial killing by macrophages polarized toward the M1-like phenotype. First, we determined the concentration range of Fe(8-hq)_3_ that could produce ROS in RAW 264.7 but would exhibit minimal cytotoxicity to the macrophage cells. Specifically, we measured the viability of macrophage-like cells treated with Fe(8-hq)_3_ from 0 to 2 µM using an MTT assay. The results showed that treating RAW 264.7 cells with Fe(8-hq)_3_ at concentrations up to 2 µM did not significantly reduce the viability of the cells when compared with the untreated control group of RAW 264.7 cells (Appendix A). Second, we quantified the amount of ROS in these macrophages using the DCFH-DA fluorescence dye to detect the ROS activity of hydroxyl, peroxyl and other free radicals inside these cells. As shown in Appendix A, the intracellular ROS level was increased in these treated cells with an increase of the Fe(8-hq)_3_ concentration in a dose-dependent manner. Finally, we examined whether the ROS generated in the presence of Fe(8-hq)_3_ was adequate to stimulate bactericidal activity of RAW 264.7 macrophages against the internalized MRSA^α^ bacteria by quantifying the number of surviving bacterial CFU within the macrophages. The results showed that the cell viability of MRSA^α^ bacteria internalized in the RAW 264.7 macrophages was significantly decreased when the concentration of Fe(8-hq)_3_ used to treat the cells was increased (Appendix A), suggesting that a bactericidal function of macrophages against intracellular bacteria is associated with the capacity of Fe(8-hq)_3_ to trigger the ROS generation in macrophages.

### 2.10. The In Vitro Evaluation of Resistance Development

Up to this point, our results appeared to suggest that the ROS signaling pathways are mainly responsible for bacterial cell death as the iron released from Fe(8-hq)_3_ can trigger the Fenton reaction. We then conjectured that Fe(8-hq)_3_ might be able to avoid resistance development that is so prevalent toward conventional antibiotics. We carried out the in vitro resistance development assay by repeatedly exposing MSSA bacteria to Fe(8-hq)_3_ for consecutive 75 passages to a sub-lethal dose in comparison with ciprofloxacin and 8-hq. At each new passage, the MIC values of Fe(8-hq)_3_, ciprofloxacin and 8-hq were determined to reveal whether any increase in resistance has developed. The repeated exposure of this bacterial strain to ciprofloxacin caused the MIC to rise by a stepwise two-fold increase from 32-fold to 256-fold of the initial MIC value on day 10, day 16, day 23 and day 39, respectively, indicating five cumulative mutational events to produce the ciprofloxacin-resistant phenotype resulting from stepwise mutations of different resistance genes. Similarly, the repeated exposure of the same bacterial strain to 8-hq resulted in four-fold, eight-fold, 16-fold, 32-fold and 64-fold increases of MIC on day 7, day 19, day 25, day 58 and day 67, respectively. The MIC value of the bacteria exposed to Fe(8-hq)_3_ increased only by four-fold on day 21, eight-fold on day 68 and then remained unchanged to the end of the experiment after 75 exposures. Furthermore, Fe(8-hq)_3_ was able to completely overcome the 8-hq resistance developed in the MSSA mutant bacteria that were exposed to 8-hq for 75 days. These results clearly indicated that the cellular targets of Fe(8-hq)_3_ were likely different and non-overlapping with those of ciprofloxacin or 8-hq. Because the cell death pathways triggered by Fe(8-hq)_3_ includes the damage to bacterial cell membrane as well as to other cellular targets due to the ROS production, it is tempting to conjecture that such cell death pathways are more irreparable than the cell death pathways triggered by ciprofloxacin or 8-hq, which renders Fe(8-hq)_3_ remarkably resilient to the development of resistance and imparts the ability to disregard the genetic mutations in MSSA^8-hqR^ (Figure 6a). Additionally, when the high-level mupirocin-resistant MRSA (MRSA^mupR^) mutant bacteria that had been exposed to mupirocin for 50 days was treated with Fe(8-hq)_3_, the MIC of Fe(8-hq)_3_ matched that of Fe(8-hq)_3_ against the wildtype MRSA and remained unchanged for the next 30 days, indicating that Fe(8-hq)_3_ has the ability to completely overcome the mupirocin resistance developed in the MRSA mutant bacteria (Figure 6b). These results suggest that the genetic mutations in the mupirocin-resistant MRSA bacteria lack the necessary defense mechanisms against the attack of Fe(8-hq)_3_, further confirming that the cellular targets of Fe(8-hq)_3_ are different and non-overlapping with those of mupirocin.

### 2.11. Checkerboard Assays

Since MRSA^α^ bacteria (ATCC BAA-44) are resistant to ciprofloxacin, oxacillin and imipenem [27], we investigated whether Fe(8-hq)_3_ could sensitize the activity of ciprofloxacin, oxacillin and imipenem using the checkerboard assay between Fe(8-hq)_3_ and the three antibiotics. First, the MICs of Fe(8-hq)_3_ was determined, followed by the measurements of the MICs of Fe(8-hq)_3_ in the presence of ciprofloxacin, oxacillin or imipenem at varying concentrations, respectively. The FIC of Fe(8-hq)_3_ was calculated by means of dividing the MIC of Fe(8-hq)_3_ in the presence of an antibiotic by the MIC of Fe(8-hq)_3_ alone. Conversely, the FIC of an antibiotic was calculated by means of dividing the MIC of the antibiotic in the presence of Fe(8-hq)_3_ by the MIC of the antibiotic alone. Hence, the FIC index χ could be obtained by summing both FIC values. When χ ≤ 0.5, the effect is referred to as synergistic; when 0.5 < χ < 4, the effect is referred to as additive; and when χ ≥ 4, the effect is referred to as antagonistic [28]. As shown in Table 2, the results showed that Fe(8-hq)_3_ exhibited a synergistic effect with both ciprofloxacin and imipenem (Appendix A). These studies suggest that a topical Fe(8-hq)_3_ ointment may be used as a combination therapy with the oral or injectable ciprofloxacin or imipenem for reducing bacterial burden in severe and/or chronic wounds with MRSA infections.

### 2.12. In Vitro Antimicrobial Effects of 2% Fe(8-hq)_3_ Ointment

Since the mechanism of action by Fe(8-hq)_3_ against SA bacteria is completely nonoverlapping with the molecular targets of other conventional antibiotics, Fe(8-hq)_3_ may therefore have potential as an alternative to both mupirocin and fusidate for treating SSTIs by MRSA. To test such hypothesis, in vitro antimicrobial activity studies were carried out in agar plates to evaluate the antimicrobial efficacy of the PEG-based ointment containing 2% Fe(8-hq)_3_ in comparison with both 2% mupirocin ointment and 2% fudidate ointment using DMSO as the negative control (vehicle) against a mupirocin- and fusidate-susceptible wildtype strain of MRSA^α^ (ATCC BAA-44), a high-level mupirocin-resistant strain of MRSA^α^ (MRSA^(mupR)^; MIC > 4 mM) and a high-level fusidate-resistant strain of MRSA^α^ (MRSA^(fusR)^; MIC ≥ 4 mM) (see Figure 6b). As shown in Figure 7a and Appendix A, treatment with either 2% antibiotic ointments or 2% Fe(8-hq)_3_ ointment in the wildtype MRSA^α^ bacteria resulted in a significant expansion of zone of inhibition in comparison to the treatment with the vehicle alone, suggesting that akin to mupirocin and fusidate, the ointment formulation of Fe(8-hq)_3_ was feasible in terms of topical deliverability of this metal complex. Specifically, the zone of inhibition generated by 2% mupirocin ointment in the wildtype of MRSA^α^ bacteria was found to have the size of 12.06 ± 0.42 cm^2^, and that generated by 2% fusidate ointment was found to have the size of 11.30 ± 0.53 cm^2^, whereas the zone of inhibition generated by 2% Fe(8-hq)_3_ ointment had a smaller size of 5.69 ± 0.12 cm^2^ when compared with those by the antibiotic ointments, reflecting the relatively lower antimicrobial potency of Fe(8-hq)_3_ in comparison with these two highly efficacious antibiotics. However, treatment with 2% Fe(8-hq)_3_ ointment in the high-level mupirocin-resistant [MRSA^(mupR)^] bacteria showed a zone of inhibition with a size (6.15 ± 0.12 cm^2^) comparable to that generated by this complex in the wildtype MRSA bacteria, i.e., 5.69 ± 0.12 cm^2^ (Figure 7b and Appendix A), although the zone of inhibition generated by 2% fusidate ointment in this strain of MRSA bacteria had a larger size of 12.30 ± 0.63 cm^2^. The latter observation revealed that there was no cross-resistance developed between mupirocin and fusidate. It should be noted that treatment with 2% mupirocin ointment in the high-level mupirocin-resistant [MRSA^(mupR)^] bacteria could still produce a zone of inhibition, albeit the size was more than ten-times smaller (i.e., 1.10 ± 0.13 cm^2^) than that produced by this antibiotic in the wildtype MRSA^α^ bacteria, indicating that mupirocin retained some antimicrobial activity against these bacteria to a certain extent but would be undoubtedly unable to eradicate the bacteria of this phenotype. Similarly, treatment with 2% Fe(8-hq)_3_ ointment in the high-level fusidate-resistant [MRSA^(fusR)^] bacteria also produced a zone of inhibition with a size (5.98 ± 0.11 cm^2^) comparable to that generated by this complex in the wildtype MRSA^α^ bacteria, i.e., 5.69 ± 0.12 cm^2^, although the zone of inhibition generated by 2% mupirocin ointment in this strain of MRSA bacteria had a larger size of 12.65 ± 0.53 cm^2^. Once again, treatment with 2% fusidate ointment against this high-level fusidate-resistant strain of MRSA^α^ [MRSA^(fusR)^] bacteria produced a noticeable zone of inhibition, but the size was about seven-times (1.53 ± 0.23 cm^2^) smaller that produced by this ointment in the wildtype bacteria. The results also indicated that no cross resistance between fusidate and mupirocin occurred in the [MRSA^(fusR)^] mutant bacteria (Figure 7c and Appendix A). Overall, our 2% Fe(8-hq)_3_ ointment could overcome the drug resistance in both [MRSA^(mupR)^] and [MRSA^(fusR)^] mutant bacteria, respectively, and exhibited nearly unchanged antimicrobial activity against the wildtype, high-level mupirocin-resistant and high-level fusidate-resistant MRSA with a zone of inhibition that was irrespective of the bacterial strain.

### 2.13. The In Vivo Validation of Therapeutic Efficacy of 2% Fe(8-hq)_3_ Ointment

The in vivo therapeutic efficacy of 2% Fe(8-hq)_3_ ointment was validated in a murine model with excisional wound infection using bioluminescent *S. aureus* (Xen36, PerkinElmer). The mice (Jax Swiss outbred) in both the treatment and control group were first inoculated with 5 × 10^6^ CFU/mL of bioluminescent *S. aureus* in the skin wound site at day 0. The IVIS images were taken in bioluminescence mode 24 h post-inoculation. A defined dose of 2% Fe(8-hq)_3_ (50 µL pre liquified) or vehicle control (2% DMSO PEG base) was topically administered to the wound site of the treatment group, while the vehicle was topically administered to the wound site of the control group each at the dose of once/day, respectively. The treatments were administered for a total of 3 days following the IVIS imaging. The wound size and in vivo bacterial burden of each anesthetized mouse were determined by measuring the bioluminescent signals of *S. aureus* (Xen36). In addition, the viable bacterial CFUs of the wound tissues were determined on day 4 post-infection. As shown in Figure 8b, 2% Fe(8-hq)_3_ ointment treatment results in decreased lesion sizes and reduction in bioluminescent signals compared with the PEG vehicle control. Compared with the control group, the treatment group reduced the bacterial burden by 99 ± 0.5%. Together with the observation that mice treated with the 2% Fe(8-hq)_3_ did not lose any weight, which indicates that the compound both significantly reduces the bacterial burden and does not result in severe side effects following topical administration, helps demonstrate the possibility of using 2% Fe(8-hq)_3_ ointment in treating skin infections by *S. aureus* (Figure 8c, Appendix A).

## 3. Materials and Methods

Chemicals and reagents used in this work were purchased from commercial suppliers. FeCl_3_, 8-hydroxyquinoline, ethyl alcohol, chloroform, HNO_3_, HCl, DMSO, 2,2′ bipyridine (bipy) and thiourea (TU) were obtained from MilliporeSigma, while the bacterial and mammalian cells were all purchased from the American Type Culture Collection (Manassas, VA, USA). Bioluminescent *Staphylococcus aureus* (Xen36) was purchased from PerkinElmer. Ciprofloxacin (≥98%), imipenem (≥98%), mupirocin (≥98%), and fusidate (≥98%) were purchased from Sigma-Aldrich (St. Louis, MS, USA). All cell culture medium and the associated supplies were obtained from Fisher Scientific (Hampton, NH, USA).

### 3.1. Synthesis of Fe(8-hq)_3_

First, 8-hydroxayquinoline (3.0 mmol) was dissolved with a 10-mL ethanolic solution in a 50-mL beaker. After iron (III) chloride (1.0 mmol) in 5 mL of ethanol 3 h. To this solution, a solution of iron (III) chloride (1.0 mmol) in 5 mL of ethanol was added and followed by 3 h-stirring of the mixed solution. The dark green precipitate was then filtered and washed with ethanol three times. The crude product was dried in a vacuum oven overnight. Elemental analysis and LC-MS measurements showed that the purity of Fe(8-hq)_3_ to be ≥98%. The product was further characterized by conventional spectroscopic techniques and X-ray powder diffraction (see the Appendix A for details).

### 3.2. Stability of Fe(8-hq)_3_ in the Bacterial Cell Culture Medium

980-μL TSB medium and 20-μL DMSO solution of Fe(8-hq)_3_ was mixed to obtain a concentration of 4 mM Fe(8-hq)_3_. The supernatant was removed by centrifugation after the mixture was first incubated at 37 °C for 48 h. The pellet was collected and dried in vacuum at room temperature for 48 h, followed by extracting the compound into 10 mL chloroform. After 4/5 of the solvent was removed by rotary evaporation, the precipitate was filtered, collected and recrystallized in CH_2_Cl_2_. The crystalline product was filtered, washed with ether and dried in vacuum at room temperature for 24 h. The ESI/LC-MS measurements show that the product that had been incubated in the cell culture medium for 48 h experienced no degradation (see Appendix A).

### 3.3. Evaluation of Minimum Inhibitory Concentrations (MICs)

We determined the MICs of 8-hydroxyquinoline and Fe(8-hq)_3_ against four strains of bacteria (see Table 1) using the standard microdilution method [29]. Briefly, bacteria (1 × 10^6^ CFU/mL) were treated with different concentrations of 8-hydroxyquinoline and Fe(8-hq)_3_, followed by transferring 200 μL of the subsequent bacterial suspension into a 96-well plate and incubating the plate at 37 °C for 24 h. The MIC value was determined as the lowest concentration where no visible bacterial growth was observed.

### 3.4. Investigation of Antibacterial Activity of 8-Hydroxyquinoline and Fe(8-hq)_3_

First, a single colony of bacteria was cultured for 24 h in TSB (5 mL: at 37 °C and 180 rpm). The bacteria suspension was diluted to 1:100 in a new medium and incubated at 180 rpm for 4 h at 37 °C to establish the density of bacteria at ca. 1 × 10^9^ CFU/mL. Then, 100 μL of the above suspension was mixed with 890 μL of medium and treated with different amounts of 8-hydroxyquinoline and Fe(8-hq)_3_ in 10 μL of DMSO. The mixture was then incubated for 24 h in an incushaker at 180 rpm at 37 °C. The number of colonies in the agar plate was counted to obtain the CFU. Such measurements were carried out in triplicate.

### 3.5. Evaluation of Biofilm Inhibition

This assay was carried out using MRSA^α^ bacteria in a well plate using a previously published procedure [30]. Bacteria at the density of 1 × 10^6^ CFU/mL were treated with varying concentrations of 8-hydroxyquinoline or Fe(8-hq)_3_. The biofilms formed by transferring 250 μL of the above suspensions into 6-well plates followed by incubation of 24 h, were carefully washed with PBS (1×) to keep them intact. The washed biofilms were then resuspended in PBS, slowly broken up and spread on agar plates. A serial dilution of each sample was first carried out in order to determine the number of bacterial colonies generated by the bacteria from the biofilms. The results were expressed as the change of CFU in reference to the negative control that was not treated with a drug.

### 3.6. Measurements of Intracellular Iron Concentrations

We used an experimental procedure previously reported by us in the literature to monitor the cellular uptake of Fe(8-hq)_3_ [14]. Briefly, bacterial cells at the density of (1 × 10^9^ CFU/mL were each treated with two different doses (i.e., 4 µM and 8 µM) of Fe(8-hq)_3_ or FeCl_3_ and incubated for 6 h. Next, 10 µL of each treated bacterial suspension was withdrawn for CFU counting in an agar plate. The rest of the bacterial suspend was each centrifuged at 25 °C and 3750 rpm for 7 min. to obtain a solid pellet that was first collected by first removing the supernatant and then washed with deionized H_2_O three times. The harvested cells were digested with concentrated nitric acid and calcined at 620 °C in air for 5 h to obtain iron oxide. The latter was treated with aqua regia and diluted to the required volume for the determination of Fe content using AAS.

### 3.7. Time–Kill Assays

We performed the time–kill assays against MRSA^α^ using an experimental procedure we reported previously in the literature [14]. 

### 3.8. Determination of Intracellular ROS Generation

The experimental procedure to measure the intracellular ROS generation triggered by Fe(8-hq)_3_ was also adopted from our previously published protocol with minor modifications [14]. MRSA^α^ bacteria were first cultured overnight, collected by centrifugation (at 3750 rpm for 7 min), and then resuspended in fresh TSB (400 μL). 100 μL of the above suspension was each mixed with 890 μL TSB, treated with 10 μL solution 8-hydroxyquinoline or Fe(8-hq)_3_ at three different concentrations. After incubated at 37 °C for 1 h with gentle agitation, bacterial pellets were harvested by centrifugation, washed with Hank’s balanced salt solution (HBSS 1×) and incubated at 37 °C with 20 μM of DCFH-DA (i.e., 2′,7′—dichlorofluorescein diacetate) for 30 min in the dark. The fluorescence intensity of the above bacterial suspension was determined using a SpectraMax M4 microplate reader. The excitation/emission wavelength used to determine fluorescence intensity was set at at 497/529 nm. 

### 3.9. Measurements of Intracellular ROS Scavenging Effect

First, we treated bacterial cells with Fe(8-hq)_3_ and thiourea (TU; 200 mM) or 2,2′-bipyridine (bipy; 250 mM), while the bacterial cells treated with Fe(8-hq)_3_ without TU or bipy were used as the comparison group to reveal the ROS scavenging effect of TU and bipy. After each group of bacterial cells was collected by centrifugation, washed with Hank’s balanced salt solution (HBBS 1×) twice, and incubated in the dark at 37 °C for 30 min with 20 μM of DCFH-DA with slight agitation, the fluorescence intensity of bacteria was measured the at 497/529 nm with a SpectraMax M4 microplate reader. 

### 3.10. Measurements of Cellular Membrane Permeabilization

The experimental procedure of visualizing bacterial membrane permeabilization was carried using a protocol described previously with modifications [31,32]. Briefly, MRSA^α^ bacteria (1 × 10^9^ CFU/mL) were incubated after being treated with varying concentrations of 8-hydroxyquinoline and Fe(8-hq)_3_ for 2 h at 180 rpm. Bacterial pellets were collected by centrifugation and resuspended in PBS with addition of 20 μM propidium iodide (PI) and allowed to incubate for 10 min in the dark. Finally, an inverted microscope (Olympus IX81, Tokyo, Japan) was used to examine the glass slides immobilized with 8 μL of each bacterial suspension. For spectrophotometric analysis, PI dye (final concentration 10 μM) was added to bacterial cells. After incubation at 37 °C in the dark for 15 min, an aliquot of 100 μL bacterial suspension was moved to a 96-well plate for determining the relative fluorescence unit (RFU) at excitation/emission wavelength of 535/617 with a SpectraMax M4 microplate reader.

### 3.11. SEM Imaging Studies of Bacterial Morphology

The visualization of cell morphology in MRSA^α^ was performed on a Quanta450 SEM instrument based on the procedure reported previously [13,14,15]. After treating the bacteria at the density of 1 × 10^9^ CFU/mL with 8-hydroxyquinoline or Fe(8-hq)_3_ each at the concentration of 32 μM, the bacteria were incubated for 2 h, followed by harvesting them by centrifugation. The solid pellet was washed with PBS (1×) and fixed with a PBS solution containing 2.5% glutaraldehyde overnight. On next day, a 1% tannic acid solution was added to the above bacterial suspension and allowed to react for 10 min. Afterwards, the sample was washed with PBS followed by ethanol solutions with increased concentrations ranging from 25%, 30%, 50%, 75% to 100%, respectively for dehydration. After drying in the air, the sample was coated with gold and imaged under SEM.

### 3.12. The In Vitro Assays of Resistance Development

We carried out the resistance development assays by successive passaging of bacteria using a procedure as described previously [14,15,33]. By serially exposing 8-hydroxyquinoline, Fe(8-hq)_3_, ciprofloxacin, and to the MSSA for 75 days, the rate of resistance development of Fe(8-hq)_3_ was compared with that of 8-hydroxyquinoline and ciprofloxacin. The change in MICs was evaluated progressively from the initial MICs of the respective drug. On each following day, bacteria were treated with a drug at the concentration of half-MIC and inoculated in fresh TSB. After incubation at 37 °C for 24 h, bacteria obtained from the growth treated with the highest concentration of each respective drug were collected for the determination of the MIC. The entire experiment lasted 75 days with the results plotted as the fold change of MIC in multiple times vs. passage.

### 3.13. Preparation of High-Level Mupirocin- and Fusidate-Resistant MRSA^α^ Strains of SA Bacteria

To evaluate the potential of mupirocin and fusidate to develop resistance, the wildtype MRSA^α^ (WT MRSA) was exposed to these two antibiotics for 50 consecutive days, respectively [13,14,15]. By monitoring the change in the bacterial cells were treated with each of them at their respective ½ × MIC, followed by inoculating the treated bacteria in fresh TSB. After incubation at 37 °C for 24 h, the bacteria grown at the highest concentration of each drug were then harvested and the new MIC value was determined. This experiment lasted for 50 days until the wildtype MRSA developed resistance to mupirocin and fusidate (MIC ≥ 4.0 mM), respectively.

### 3.14. Evaluation of Cytotoxicity of Fe(8-hq)_3_ in RAW 264.7 Cells

We performed MTT viability assay to determine the cytotoxicity of Fe(8-hq)_3_ to RAW 264.7. In a 96-well plate bacterial cells were seeded with DMEM high-glucose medium to the density of 4 × 10^5^ cells/well. The plate was incubated at 37 °C in a 5% CO_2_ atmosphere for 24 h, followed by treating the cells with varying amounts of Fe(8-hq)_3_ in 100 μL of fresh medium and incubating them for 24 h. To each well, 10 μL of MTT reagent was added after the medium was replaced with fresh one. After incubation for 2 h at 37 °C, to each well, 100 μL of detergent reagent was added. After being kept in the dark for 2 h at 37 °C, the plate was read at 570 nm with a SpectraMax M4 microplate reader. 

### 3.15. Measurements of Intracellular ROS Generation in Macrophage-like Cells

The total level of intracellular ROS in macrophage-like cells (RAW 264.7) was measured using the method as reported previously [26]. Briefly, RAW 264.7 cells were incubated with DCFH-DA dye after being treated with varying concentration of Fe(8-hq)_3_ for 24 h, and fluorescence intensity was evaluated using a microplate reader (SpectraMax^®^ M4).

### 3.16. Quantitative Measurements of Bactericidal Activity of Macrophages

We evaluated the effect of Fe(8-hq)_3_ to trigger bactericidal activity of macrophage-like RAW 264.7 cells against internalized *S. aureus* bacteria using a procedure as described previously [26,34]. In a 24-well plate, each well was seeded with RAW 264.7 cells at a density of 4 × 10^4^ cells per well with the DMEM medium. Cells in each well were washed with PBS and selected concentration of the Fe(8-hq)_3_ (0.5–2 µM) in 400 μL fresh medium and incubated for 24 h. Overnight bacteria culture MRSA^α^ was subculture at 1:100 dilutions in 5 mL of TSB medium and incubated at 37 °C with shaking at 180 rpm for 4 h to obtain 1 × 10^9^ CFU/mL. Then, 32 µL of subculture bacteria were diluted in 5 mL DMEM basal media (no serum and no antibiotics). After washing the cell, 500 µL of diluted bacterial suspension was transferred to each well and incubate for 60 min. After the supernatant was removed, the cells were washed with PBS, followed by addition of 500 µL of DMEM basal media with 10% FBS and 50 µg/mL gentamicin to each well. Again, the plate was incubated at 37 °C, 5% CO_2_ for 24 h. After incubation supernatant was removed by aspiration and cells were washed with PBS. Cells were lysed with sterilized 1% saponin in water per well by incubating for 10 min at room temperature followed by vigorous pipetting and 50 µL of undiluted lysate was spread on the agar for colony counting. All experiments were conducted in triplicate.

### 3.17. Checkerboard Assays

The checkerboard assays were carried out in a 96-well plate using a procedure as reported previously [14,35]. Two antibiotics, i.e., oxacillin and imipenem tested for a potential synergistic effect with Fe(8-hq)_3_. The antibiotics were serially diluted 2-fold each time and placed in wells along the row-axis. Similarly, the wells along the column-axis contained Fe(8-hq)_3_ solutions that were also serially diluted by 2-fold each time, creating a checkerboard matrix. Therefore, each well contained a combination of two drugs with varying concentrations. After inoculating the wells with MRSA^α^ and adjusting the volume of each well to 100 µL to obtain a bacterial density reached of 1  ×  10^6^/mL, the well-plate was kept at 37 °C for 20 h before visual inspection was carried out to obtain MICs. The fractional inhibitory concentration (FIC) of Fe(8-hq)_3_ was defined as the quotient of the MIC of Fe(8-hq)_3_ in combination with an antibiotic divided by the MIC of Fe(8-hq)_3_ alone. Conversely, the FIC of an antibiotic was defined as the quotient of the MIC of the antibiotic in the presence of Fe(8-hq)_3_ divided by the MIC of the antibiotic alone. The FIC index (χ) is the sum of two FIC values. When χ ≤ 0.5, the effect is referred to as synergistic; when 0.5 < χ < 4, the effect is referred to as additive; and when (χ ≥ 4), the effect is referred to as antagonistic [28].

### 3.18. Preparation of Topical Ointments Containing 2% Mupirocin, 2% Fusidate or 2% Fe(8-hq)_3_, Respectively

The ointment base of polyethylene glycol (PEG) was prepared in accordance with the U.S. Pharmacopeia and The National Formulary (USP 24-NF 19) [36]. First, 50 mg of mupirocin, fusidate or Fe(8-hq) dissolved in 250 µL DMSO was immediately added to 2.5 g of PEG ointment base in the liquid state by keeping the solution at 60 °C to generate a homogenous dispersion containing 2% mupirocin, 2% fusidate or 2% Fe(8-hq)_3_. These solution mixtures were then allowed to cool to room temperature to solidify (see Appendix A).

### 3.19. Validation of In Vitro Antimicrobial Efficacy of the Ointment

The in vitro antimicrobial efficacy of the ointment was validated by measuring zones of inhibition in MRSA bacteria using a published procedure with modifications [36]. First, MRSA bacteria in 100 µL at the density of 1 × 10^8^ CFU/mL of were spread on TSA well-plates. After drying the plates for 10 min, an aliquot of ointment (40 µL) was added to the center of each agar plate. After incubation at 37 °C for 24 h, the size of inhibition zone in each agar plate was measured and the images were processed using ImageJ.

### 3.20. Excisional Murine Skin Wound Infection Model with S. aureus

All animal procedures were performed in accordance with the Guide for Care and Use of Laboratory Animals and approved by the Institutional Animal Care & Use Committee (IACUC) of the Department of Pharmaceutical Sciences, College of Pharmacy, Northeast Ohio Medical University. Jax Swiss outbred mice (male and female mice, 8–12 weeks old) used as in vivo animal models for our studies were obtained from a commercial source. Before the local wound infection was created, anesthesia was induced by inhalation of 3–5% isoflurane gas and maintained by continued inhalation of 2–3% isoflurane. The mice were then shaved, and the planned biopsy/infection site was swabbed with 2% povidone-iodine followed by 70% iodine. A circular wound with full-thickness was inflicted on the dorsal surface of the animal using a sterile biopsy punch of 5 mm. The wound was coated with a Tegaderm^®^ dressing to form a transparent and semipermeable layer. Next, 40 μL of bioluminescent *S. aureus* (Xen36, 5 × 10^6^ CFU/mL) suspended in PBS was introduced into the wound site under the dressing. On day 1 of post-infection, IVIS images were taken in bioluminescence mode with auto exposure settings. A defined dose of 2% Fe(8-hq)_3_ (50 µL pre-liquified) or vehicle control (2% DMSO PEG base) was applied to the wounds of either the treatment group or the vehicle control group every 24 h. The treatments were administered for a total of 3 days following the IVIS imaging. At day 4 post-infection, animals were euthanized, and the wounded skin was excised with a 6 mm biopsy punch and homogenized for bacterial CFU counting on LB agar plates. After 24 h, colonies were counted, and the plates were imaged using the IVIS system. Non-luminescent colonies were subtracted from the colony count to establish the bacterial load that had been present on the biopsied tissue.

### 3.21. Statistical Analysis

Details of statistical analysis used in this work are similar to those used in our previously published studies and can be found somewhere else [14]. Briefly, statistical analysis was performed using GraphPad Prism version 8.0 software. A two-tailed unpaired *t*-test was used to determine statistical significance between two groups. A statistical significance among multiple groups was analyzed using one-way ANOVA followed by Holm–Sidak comparisons test. For all analyses, *p*-value of less than 0.05 was considered to be statistically significant. Data were presented as mean ± standard deviation (mean ± s.d).

## 4. Conclusions

In conclusion, the antimicrobial mode of action of 8-hq stems from its ability to chelate metal ions such as Mn^2+^, Zn^2+^ and Cu^2+^ in the bacterial cell to disrupt the cellular metal homeostasis. Complexation of Fe(III) by 8-hq adds another mode of action of bacterial killing by harnessing the cytotoxicity of iron. Apparently, the *D*3 molecular symmetry in Fe(8-hq)_3_ not only nullifies the overall molecular dipole moment of this complex but also conceals the ionic character of Fe(III), granting the complex “greaseball-like” characteristics. Hence, the complexation of Fe(III) with this judiciously chosen ligand forms an electrically neutral and highly lipophilic molecule of Fe(8-hq)_3_ that can greatly facilitate the transport of iron across the bacterial cell membrane. As a result, the antimicrobial activity of Fe(8-hq)_3_ against MRSA and MSSA is significantly enhanced. The measured intracellular ROS production triggered by Fe(8-hq)_3_, but not by 8-hq, is therefore attributable to the redox activity of iron in relation to its ability to catalyze the Fenton reaction. The observed cell membrane damage revealed by SEM imaging and PI dye fluorescence studies in the cells treated with Fe(8-hq)_3_, but not with 8-hq, is fully consistent with the intracellular ROS production induced by Fe(8-hq)_3_. Interestingly, intracellular ROS production in macrophages can readily polarize such mammalian cells toward the M1-like phenotype to exhibit a bactericidal effect against the bacteria internalized in macrophages. Because the cellular targets of Fe(8-hq)_3_ are different and non-overlapping with those of ciprofloxacin or of 8-hq itself, Fe(8-hq)_3_ shows remarkable resilience to the development of resistance. The synergistic effect observed between Fe(8-hq)_3_ and either ciprofloxacin or imipenem offers potential to use Fe(8-hq)_3_ to potentiate either one of these two antibiotics. Since the citrate salt of 8-hq is already an FDA-approved drug for treating vaginal infections, Fe(8-hq)_3_ may prove to be suitable for topical applications either as a monotherapy or as a combination therapy for MRSA infections. This work shows that the lipophilic Fe(III)-chelating agent 8-hq can act as an effective ionophore to transport the highly charged Fe^3+^ ion across the bacterial cell membrane to afford significant enhancement of antimicrobial activity of 8-hq against SA by harnessing the cytotoxicity of iron. With the emergence of multi-drug resistant MRSA, the need for nonconventional topical antimicrobial drugs has never been so great. Fe(8-hq)_3_ appears to be suitable to fill such gap.

## Figures and Tables

**Figure 1 antibiotics-12-00886-f001:**
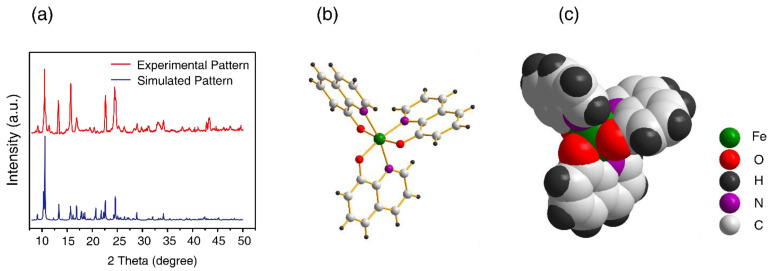
PXRD pattern of the isolated product (**a**) and X-ray structure of Fe(8-hq)_3_ with the stick-and-ball presentation (**b**) and with the space-filling presentation (**c**).

**Figure 2 antibiotics-12-00886-f002:**
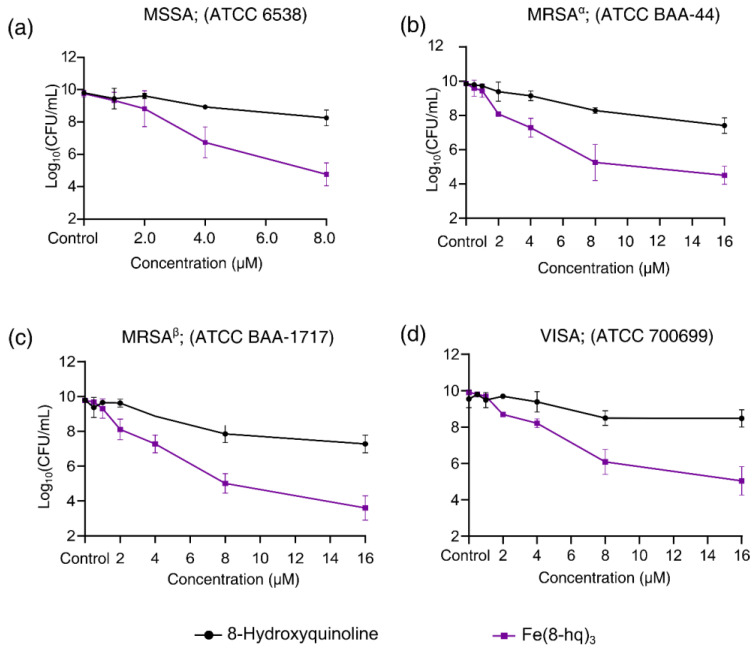
Inhibitory effects of Fe(8-hq)_3_ in comparison with molar equivalents of 8-hq against MSSA (**a**), against MRSA^α^ (**b**), against MRSA^β^ (**c**) and against VISA (**d**).

**Figure 3 antibiotics-12-00886-f003:**
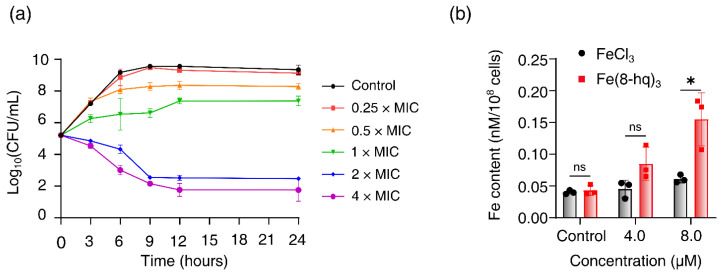
Time–kill kinetics of Fe(8-hq)_3_ against MRSA^α^ after 24-h incubation with different concentrations of Fe(8-hq)_3_ (**a**) and cellular uptake of Fe(8-hq)_3_ in MRSA^α^ as represented by the Fe content of the cell lysate (**b**) (data presented as mean ± s.d, *n* = 3 replicates; * *p* < 0.05, ns = not significant).

**Figure 4 antibiotics-12-00886-f004:**
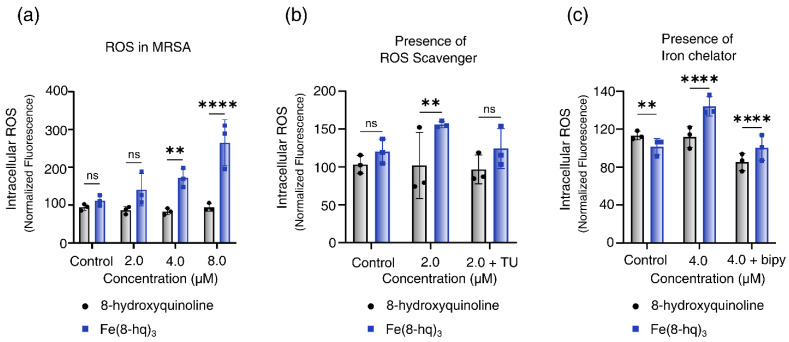
Relative yields of intracellular ROS generation in MRSA^α^ bacterial cells treated with Fe(8-hq)_3_ in comparison with molar equivalents of 8-hq (**a**), the intracellular ROS generation in MRSA^α^ bacterial cells inhibited by TU (**b**) and by bipy (**c**) (mean ± s.d, *n* = 3 replicates; ** *p* < 0.01, **** *p* < 0.0001 and ns = not significant).

**Figure 5 antibiotics-12-00886-f005:**
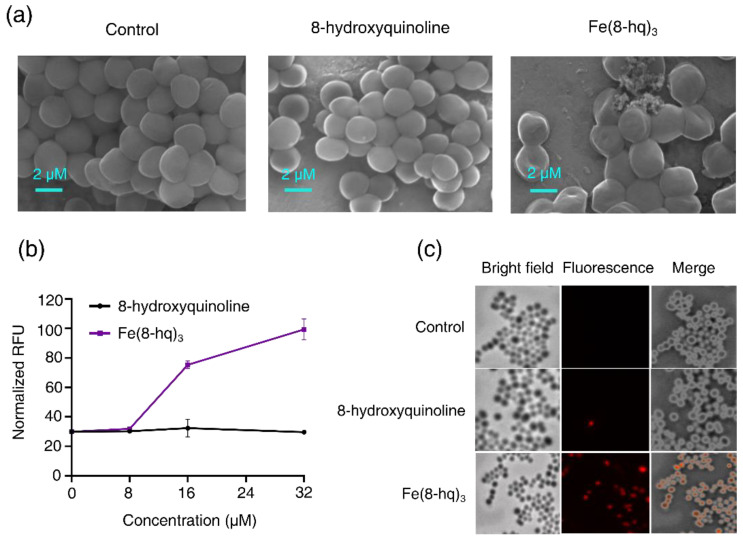
SEM images of MRSA^α^ treated with 8-hq and Fe(8-hq)_3_ both at the concentration of 32 μM for 2 h (**a**) and the change in PI fluorescence in MRSA^α^ treated with 8-hq and Fe(8-hq)_3_ (mean ± s.d, *n* = 3) (**b**) and microscopic images of PI in MRSA^α^ treated with 8-hq and Fe(8-hq)_3_ (both at 32 μM) with differential interference contrast (DIC) microscopic images (left), fluorescence images (middle) and merged images of MRSA^α^ (right) (**c**).

**Figure 6 antibiotics-12-00886-f006:**
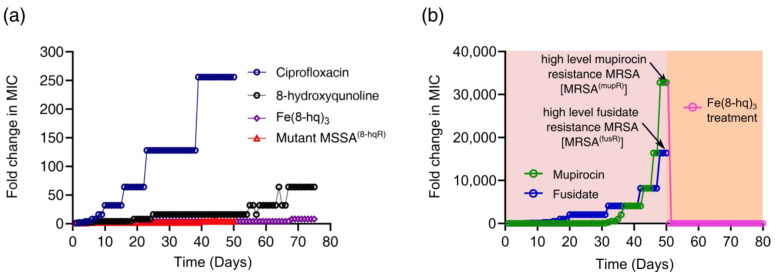
Drug resistance development of Fe(8-hq)_3_ vs. ciprofloxacin and 8-hq in MSSA and the overcome of drug resistance the in mutant MSSA^8-hqR^ (a strain resistant to 8-hq) bacteria by Fe(8-hq)_3_ without developing Fe(8-hq)_3_ resistance after 50 days of treatment with Fe(8-hq)_3_ (**a**), and resistance development profile of mupirocin and fusidate in MRSA^α^ and the overcome of high-level mupirocin resistance mutant bacteria (MRSA^mupR^) by Fe(8-hq)_3_ (**b**).

**Figure 7 antibiotics-12-00886-f007:**
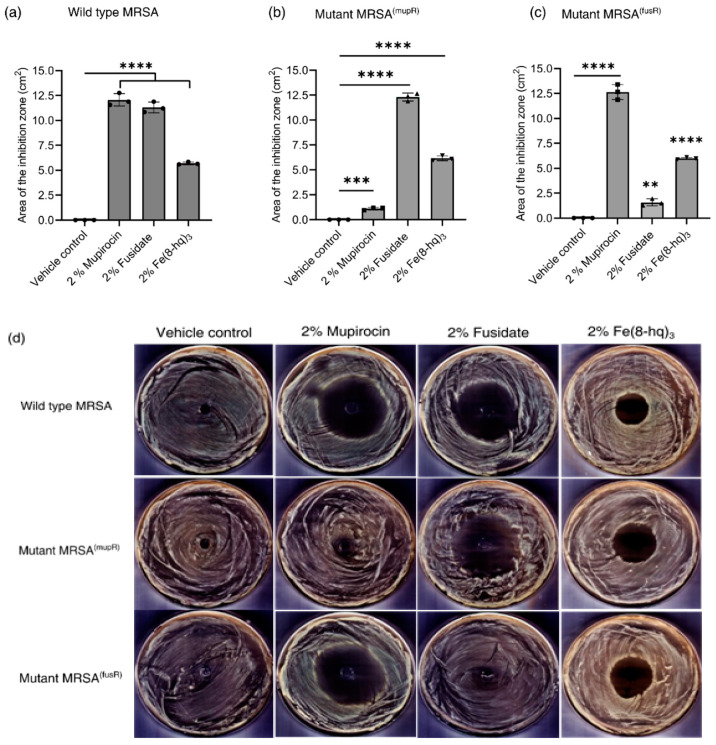
Results of growth inhibition measurements (**a**–**d**). The average zones of inhibition of PEG-based ointments containing the 2% mupirocin, 2% fusidate and 2% Fe(8-hq)_3_ towards MRSA strain with wildtype MRSA^α^ (**a**), high-level mupirocin resistant MRSA^α^; MRSA ^(mupR)^ (**b**), high-level fusidate resistant MRSA^α^; MRSA^(fusR)^ (**c**) and representative images of antimicrobial zone of growth inhibition of different bacterial strains (**d**). (mean ± s.d, *n* = 3 replicates; ** *p* < 0.01, *** *p* < 0.001, and **** *p* < 0.0001).

**Figure 8 antibiotics-12-00886-f008:**
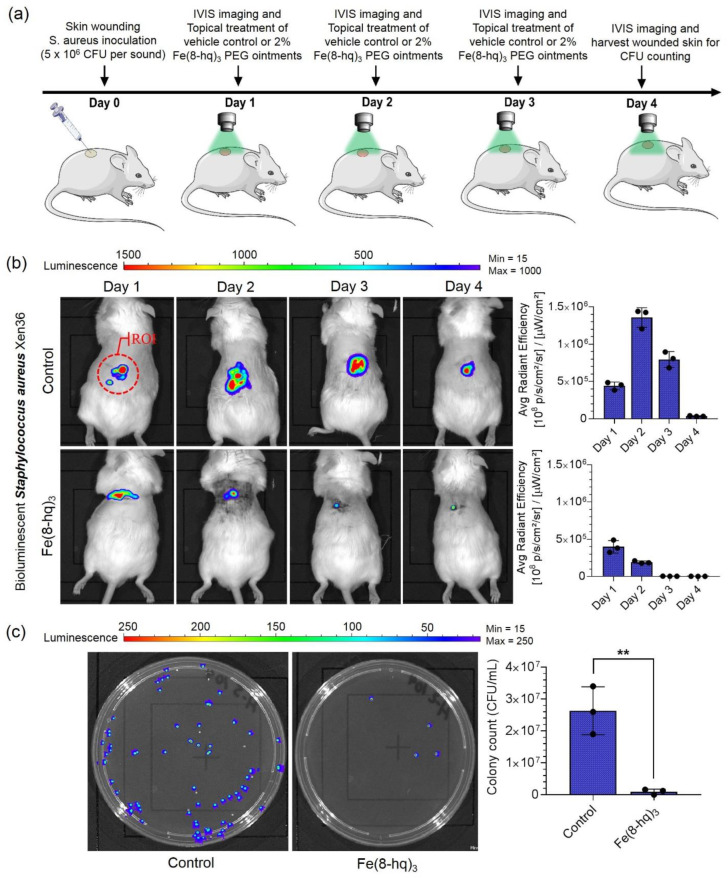
Treatment outcomes of in vivo wound infections by *S. aureus* (Xen36) in a murine model with the excisional wound. Overview of the in vivo experiments (**a**), results of in vivo bioluminescent imaging studies of mice with ROI measurements (**b**), and representative images of CFU enumeration result against *S. aureus* (**c**) (mean ± s.d, *n* = 3 mice per group; ** *p* < 0.01).

**Table 1 antibiotics-12-00886-t001:** MIC values of Fe(8-hq)_3_ against four different strains of SA bacteria.

Bacteria Stains	MIC (in µM)	
	Fe(8-hq)_3_	8-hq
MSSA (ATCC 6538)	4 µM	16 µM
MRSA^α^ (ATCC BAA-44)	4 µM	16 µM
MRSA^β^ (USA 300, ATCC BAA-1717)	4 µM	32 µM
VISA (ATCC 700699)	4 µM	16 µM

**Table 2 antibiotics-12-00886-t002:** The determination of MIC values and FIC indexes of Fe(8-hq)_3_ and ciprofloxacin (a), Fe(8-hq)_3_ and imipenem (b) against MRSA^α^.

	MICs against MRSA^α^ (ATCC BAA-44)				FIC Index
(a)	Ciprofloxacin only	Ciprofloxacin with Fe(8-hq)_3_	Fe(8-hq)_3_ only	Fe(8-hq)_3_ with Ciprofloxacin	
	48.0 µM	6.0 µM	4.0 µM	1.0 µM	0.375
(b)	MICs against MRSA^α^ (ATCC BAA-44)				FIC Index
	Imipenem only	Imipenem with Fe(8-hq)_3_	Fe(8-hq)_3_ only	Fe(8-hq)_3_ with Imipenem	
	50.0 µM	6.25µM	4.0 µM	1.0 µM	0.375

## Data Availability

All data generated or analyzed during this study are included in this published article and its Appendix A.

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
