# Peer review of "Harnessing the Dual Antimicrobial Mechanism of Action with Fe(8-Hydroxyquinoline)_3_ to Develop a Topical Ointment for Mupirocin-Resistant MRSA Infections"

_antibiotics, 2023, doi:10.3390/antibiotics12050886_

Round 1
Reviewer 1 Report
This is a very interesting manuscript demonstrating the antimicrobial activities of Fe(8-hydroxyquinoline)3 against MRSA, MSSA, and VISA, but not on the MRSA, MSSA, or VISA infection.
Why Fe is selected and focused, not Ca, Cu, or etc?
The formation of Fe(8-hq)3 seems to be temporary, and easily dissociated when exposed to cell culture medium. This phenomena is hinting that the bioavailability of Fe(8-hq)3 could be very short and lacking of specificity. Any recommendation to improve it? How to ensure that the Fe3+ will be released after entering the targeted microbial?
Indeed, only the in vivo part demonstrated the efficacy of Fe(8-hq)3 in treating SSTI, but not involving the MRSA. This seems like not align with the manuscript title. To be precise, there is not MRSA infection study in this manuscript.
Please discuss more in the Results & Discussions section. Majority of the writings are elaboration and description of acquired results. Indeed, MRSA is not a new topic, should be no problem for the authors to compare their findings with those published.
Table1: Please revise and confirm the heading. There are two columns of values sharing only one heading.
Figure2: Missing of labels for statistic significancy, despite mentioned in figure legend.
L599: ...layer. And. Next,...
References: Please italicize all the species name in the publication title.
Author Response
Reviewer 1:
Comments to the Authors
This is a very interesting manuscript demonstrating the antimicrobial activities of Fe(8-hydroxyquinoline)3 against MRSA, MSSA, and VISA, but not on the MRSA, MSSA, or VISA infection.
Why Fe is selected and focused, not Ca, Cu, or etc?
Answer: it is well established that the antimicrobial activity of 8-hydroxyquinoline (8-Hq) stems from its ability to chelate Mn(II), Zn(II) and Cu(II). On the other hand, Fe(II) is known to trigger the Fenton catalytic reaction. In this publication, we used 8-Hq to deliver Fe(III) into the bacterial cell and concomitantly remove Mn(II), Zn(II) and Cu(II) to derive a “dual mechanism”. In other words, use of Fe is more advantageous than any of the above metals.
The formation of Fe(8-hq)3 seems to be temporary, and easily dissociated when exposed to cell culture medium. This phenomena is hinting that the bioavailability of Fe(8-hq)3 could be very short and lacking of specificity. Any recommendation to improve it? How to ensure that the Fe3+ will be released after entering the targeted microbial?
Answer: our studies using ESI/LC-MS on the recovered product Fe(8-hq)3 after incubating at 37 ºC and 180 rpm for 48 hours with the cell culture medium showed that this metal complex is stable and non-dissociative for as long as the bioassays last (<24 hours). These results are presented in the Supporting Information as Figure S4.
Indeed, only the in vivo part demonstrated the efficacy of Fe(8-hq)3 in treating SSTI, but not involving the MRSA. This seems like not align with the manuscript title. To be precise, there is not MRSA infection study in this manuscript.
Answer: the in vivo animal studies requires the use of bioluminescent strain of SA bacteria in order to generate images. Unfortunately, the bioluminescent strain of MRSA is not commercially available. Therefore, only the regular SA infection model was used in our studies. However, based on all the in vitro data, particularly the data presented in Figure 7, we are very confident that our topical cream is effective against MRSA.
Please discuss more in the Results & Discussions section. Majority of the writings are elaboration and description of acquired results. Indeed, MRSA is not a new topic, should be no problem for the authors to compare their findings with those published.
Answer: in general, we discussed the significance, relationship between structure and antimicrobial activity of this metal complex under each sub-title after the research findings are presented. As for comparing our findings with those in the literature is concerned, the research is lacking in the development of topical creams for treat MRSA SSTIs. Therefore, we are constrained to do more than we have presented in the paper.
Table1: Please revise and confirm the heading. There are two columns of values sharing only one heading.
Answer: we’d like to thank this review for catching this error. Two headings in Table were advertently cut off in our previous manuscript. They are now added (highlighted in yellow).
Figure2: Missing of labels for statistic significancy, despite mentioned in figure legend.
Answer: this figure contains values with standard deviations for each data point. No statistical analysis was done for these data. Therefore, we removed the statement on p values from the figure legend.
L599: ...layer. And. Next,...
Answer: again, many thanks for catching this error. The change is made in yellow highlight.
References: Please italicize all the species name in the publication title.
Answer: yes, we have done that now.
Reviewer 2 Report
The present manuscript entitled “Harnessing the Dual Antimicrobial Mechanism of Action with Fe(8-hydroxyquinoline) to Develop a Topical Ointment for 3 Mupirocin-Resistant MRSA Infections” by Abeydeera et al., describes the 8-Hydroxyquinoline (8-hq) the complex formed between Fe(III) and 8-hq, can readily transport Fe(III) across the bacterial cell membrane and deliver iron into the bacterial cell, thus harnessing a dual antimicrobial mechanism of action that combines the bactericidal activity of iron with the metal chelating effect of 8-hq to kill bacteria. Furthermore, Fe(8-hq)3 exhibits a synergistic effect with ciprofloxacin and imipenem, showing potential for combination therapies of topical and systemic drugs for more serious MRSA infections. The authors report an interesting work. The objective and justification of the work are clear. I appreciate the authors for their good presented work. Therefore, I recommend it for publication. However, few Minor issues are detailed below which need to be addressed before its final acceptance in the Antibiotics.
I advise the authors to take the following minor points into account while revising their manuscript.
Comment 1: There are some typographical and grammatical errors in the manuscript text, so the authors need to correct them in the revised manuscript. For. E.g. Line 13, “8-Hydroxyquinoline (8-hq) exhibits potent antimicrobial activity (SA) against Staphylococcus aureus (SA) bacteria” should be “8-Hydroxyquinoline (8-hq) exhibits potent antimicrobial activity against Staphylococcus aureus (SA) bacteria”
Comment 2: English needs to be a little improved, as there are some misused conjunctions and technical flaws. So it needs to be corrected in the manuscript.
Comment 3: The Abstract needs to be revised, include the performed characterization techniques in the abstract.
Comment 4: At the end of the Introduction, the main objectives of this study should be clearly and detailed presented. So slightly enhance the end of the Introduction section.
Comment 5: Authors performed characterization techniques PXRD, UV-Vis, and FT-IR for the as-synthesized product. However I did not observed any instrumental details (such as model number, manufacturer, and place of origin) of the performed characterization techniques, also there is no peak positions description in the manuscript text as well as in the supplementary file. So I would suggest the authors add this information in the revised manuscript.
Comment 6: If possible please include the graphical abstract for their well-presented work.
Comment 7: The homogeneity of the reference section needs to be maintained. In references, the bacteria names should be in italics. So please check and revise the references.
Moderate editing of English language is required to the manuscript.
Author Response
Reviewer 2:
Comments to the Authors
The present manuscript entitled “Harnessing the Dual Antimicrobial Mechanism of Action with Fe(8-hydroxyquinoline) to Develop a Topical Ointment for 3 Mupirocin-Resistant MRSA Infections” by Abeydeera et al., describes the 8-Hydroxyquinoline (8-hq) the complex formed between Fe(III) and 8-hq, can readily transport Fe(III) across the bacterial cell membrane and deliver iron into the bacterial cell, thus harnessing a dual antimicrobial mechanism of action that combines the bactericidal activity of iron with the metal chelating effect of 8-hq to kill bacteria. Furthermore, Fe(8-hq)3 exhibits a synergistic effect with ciprofloxacin and imipenem, showing potential for combination therapies of topical and systemic drugs for more serious MRSA infections. The authors report an interesting work. The objective and justification of the work are clear. I appreciate the authors for their good presented work. Therefore, I recommend it for publication. However, few Minor issues are detailed below which need to be addressed before its final acceptance in the Antibiotics.
I advise the authors to take the following minor points into account while revising their manuscript.
Comment 1: There are some typographical and grammatical errors in the manuscript text, so the authors need to correct them in the revised manuscript. For. E.g. Line 13, “8-Hydroxyquinoline (8-hq) exhibits potent antimicrobial activity (SA) against Staphylococcus aureus (SA) bacteria” should be “8-Hydroxyquinoline (8-hq) exhibits potent antimicrobial activity against Staphylococcus aureus (SA) bacteria”
Answer: indeed, we have found some more typos and grammatical errors after reading this reviewer’s comment. We have made all corrections when necessary (given in yellow highlight)
Comment 2: English needs to be a little improved, as there are some misused conjunctions and technical flaws. So it needs to be corrected in the manuscript.
Answer: Dr. Bogdan Benin, the second author of this article is a native English speaker and has published profusely in research articles. He carefully went through the manuscript and eliminated most, if not all, the grammatical errors.
Comment 3: The Abstract needs to be revised, include the performed characterization techniques in the abstract.
Answer: we consider the characterization to be peripheral studies necessary to confirm the identity and purity of the drug complex used in the studies or bioassays to reveal the activity, not the innovative part of the studies. In order to keep the abstract at a reasonable length, we did not add the peripheral information into the abstract.
Comment 4: At the end of the Introduction, the main objectives of this study should be clearly and detailed presented. So slightly enhance the end of the Introduction section.
Answer: we’d like to thank the reviewer for this suggestion and revised the pertinent part of the introduction as following (Line 58 – Line 62):
“Our objectives of this study are to develop novel topical antimicrobial agents for treating mupirocin-resistant and fusidate-resistant SSTIs by MRSA, particularly by the mupirocin-resistant and fusidate-resistant MRSA. We focus on the design and synthesis of metal-complexes that can target cellular and molecular components that are different from those targeted by conventional antibiotics”
Comment 5: Authors performed characterization techniques PXRD, UV-Vis, and FT-IR for the as-synthesized product. However I did not observed any instrumental details (such as model number, manufacturer, and place of origin) of the performed characterization techniques, also there is no peak positions description in the manuscript text as well as in the supplementary file. So I would suggest the authors add this information in the revised manuscript.
Answer: please see our answer to Comment 3. We’d like to point out that all characterization was carried out using the conventional and standard instruments. Since the focus of this article is not the synthesis and characterization of novel complexes but a known compound in the literature. The interested parties can use pretty much any makes of their own instruments to perform characterization work and the reproducibility is not an issue.
Comment 6: If possible please include the graphical abstract for their well-presented work.
Answer: thanks for the suggestion. We have uploaded a graphical abstract as part of our submission to this journal.
Comment 7: The homogeneity of the reference section needs to be maintained. In references, the bacteria names should be in italics. So please check and revise the references.
Answer: again, thanks for the suggestion. We have made efforts to ensure the references are in the same format and bacterial strains are in italics.
Reviewer 3 Report
The suggestions are Attached for the authors

Author Response
Reviewer 3:
Comments to the Authors
* In general, the article: Harnessing the Dual Antimicrobial Mechanism of Action with Fe(8-hydroxyquinoline)3 to Develop a Topical Ointment for Mupirocin-Resistant MRSA Infections needs several adjustments, described below:
* In title: Please put the name of the species of bacteria: Staphylococcus aureus
Answer: we have made the change as suggested.
* Introduction: Put who described each species of bacteria when citing the scientific name of the specie for the first time.
Answer: we have trouble finding this type of information in the literature. Please suggest the best place we should get started (a textbook or a review would be greatly appreciated)
Remember the rules of scientific naming. Do this with all the species mentioned in first time.
* Materials and Methods The cells were cultured in American Type Culture Collection. Authors have given no evidence of testing cultures for contamination (bacterial or fungi, etc). Moreover authors have not made attempt to detect mycoplasma contamination in cell cultures. Please include all stringency to avoid such contamination.
Answer: the cell culture techniques we adopted are very standard methods used by microbiology labs across the globe. We have been used these techniques in our research for over a decade and produced more than a dozen of publications. Specifically, mycoplasma contamination in cell cultures is not common if the growth mediums and apparatus are sterilized (as we routinely do).
* Excisional murine skin wound infection model with S. aureus. What is the registration number of the ethics committee?
Simply saying that All animal procedures were performed in accordance with the Guide for Care and Use of Laboratory An-589 imals and approved by the Institutional Animal Care & Use Committee (IACUC) is NOT enough in animal ethical terms. PLEASE ATTACH THE DOCUMENT PROOFING THAT THE RESEARCH WAS ACCEPTED WITH THE REGISTRATION NUMBER.
Answer: we have now included this information in the revised manuscript (highlighted in yellow).
*Results, Figures 7 and 8b: Please, in the figures it is essential to add a small bar drawn on the image, whose length indicates a measurement to be used as a reference for the size of the cells/objects in the image.
Answer: we checked other research articles that publish disk diffusion images similar to those in our Figures 7 and 8b. It turned our that it is not a common practice to add scalar bars to these images that are obtained with standard agar plates
*Results, Figure 8b: What statistical test is used?
Please put the name in the caption. *Statistical analysis. How did you come to the conclusion Statistical analysis without first having done the test to verify whether the data have a normal distribution or not. What test was performed to verify the normality of the data? How many biological replicates were performed? Please include all this information in the topic: Statistics
Answer: we have added the following to the end of “Materials and Methods”:
Statistical analysis. Details of statistical analysis used in this work are similar to those used in our previously published studies and can be found somewhere else [14]. Briefly, statistical analysis was performed using GraphPad Prism version 8.0 software. A two-tailed unpaired t-test was used to determine statistical significance between two groups. A statistical significance among multiple groups was analyzed using One-way ANOVA followed by Holm-Sidak comparisons test. For all analyses, p-value of less than 0.05 was considered to be statistically significant. Data were presented as mean ± standard deviation (mean ± s.d).
Round 2
Reviewer 1 Report
Fenton catalytic reaction is refers to the reaction of Fe2+ with hydrogen peroxide (the ROS in the manuscript) and the product would be the highly toxic hydroxyl radical. When the ROS increase with the increase concentration of Fe(8-hq)3, that might be hinting the Fenton Catalytic Reaction is not present. The authors may prove the present of Fenton reaction by determining the changes of Fe2+ (if the Fe(8-hq)3 is dissociated and oxidized) or the amount of hydroxyl radicals.
If Fenton Catalytic Reaction is present, the authors should be worry about the toxicity of the hydroxyl radicals that is building up, which may induce inflammation to the applied wound. Since the wound is recovering in in vivo model, again, Fenton catalytic reaction might be absent.
Regarding the action mechanism of Fe(8-hq)3 in killing the SA, more studies need to be done, but the authors shall not make any hypothesis in discussion without basis. In addition, a hypothesize claim shall not be included in keyword, which is misleading the readers. Thus, I suggest to remove the "Fenton Reaction" from keyword.
Similar phenomena was found in the ESI-MS data where the authors claimed that Fe(8-hq)3 is stable after incubating in cell culture medium. If stable, how to get Fe2+ to initiate the Fenton Catalytic Reaction?
In method (L413), the authors mentioned the stability test was done on bacteria culture medium, but in supplementary and results & discussions, it become cell culture medium. Please clarify.
Other than these, I am satisfied with the responses provided on my earlier comments. Thank you for the clarification and manuscript improvement.
Author Response
On behalf of all authors, I’d like to that this reviewer for his(her) very careful review and constructive criticism in round 2 of reviewing this manuscript. We hope that we have address all his(her) concerns now.
Reviewer 2:
Comments to the Authors
Fenton catalytic reaction is refers to the reaction of Fe2+ with hydrogen peroxide (the ROS in the manuscript) and the product would be the highly toxic hydroxyl radical. When the ROS increase with the increase concentration of Fe(8-hq)3, that might be hinting the Fenton Catalytic Reaction is not present. The authors may prove the present of Fenton reaction by determining the changes of Fe2+ (if the Fe(8-hq)3 is dissociated and oxidized) or the amount of hydroxyl radicals.
Answer: there are two separate experiments we used to prove that Fe released from Fe(8-hq)3 is responsible to the ROS production by the Fenton catalytic reaction. First, in Figure 4(a), we presented results to show that the intracellular ROS concentration increases with increasing concentration of Fe(8-hq)3 but not with 8-hq, showing the first piece of evidence that Fe might have something to do the ROS production since 8-hq is shown to have nothing to do with this. Second, in Figure 4(b), we used thiourea (TU) - a known ROS scavenger (i.e., it can absorb or neutralize ROS) to show that the fluorescent signals (yes, the assay use a fluorescent dye) we measured in Figure 4(a) as well as in Figure 4(b) is indeed caused by ROS not some kind of artifacts. However, this experiment does not answer the question about which species is causing the ROS production. We then used 2,2ᶦ-bipyridine (bipy) to prove that Fe2+ is responsible for the ROS production as shown in Figure 4(c). The chelating agent bipy can form a coordinatively saturated Fe2+ complex (its structure is shown below), thus completely passivate the catalytic activity of Fe2+ in the Fenton reaction. This is a well-known experimental technique used by cellular biologists to probe if Fe2+ causes the ROS production in bacterial and mammalian cells.
Molecular structure of FeII(2,2’bipy)3]2+
If Fenton Catalytic Reaction is present, the authors should be worry about the toxicity of the hydroxyl radicals that is building up, which may induce inflammation to the applied wound. Since the wound is recovering in in vivo model, again, Fenton catalytic reaction might be absent.
Answer: at the concentration used in the topical cream (2%), the skin toxicity caused by the ROS production should be negligible (i.e., should be even less than caused by some antiseptic such as povidone-iodine®). As a matter of fact, at the time of skin infection or injury, the human body will secrete a small amount of ROS to trigger macrophage cells towards the polarization of M1-like phenotype which is proinflammatory microbiocidal to prevent infection from developing into more serious conditions. We presented results in Figure S10(a) and (b) to show the ROS production triggered by Fe2+ released from Fe(8-hq)3 into these immune cells can also stimulate the polarization of macrophage cells to do just that. Of course, prolonged inflammation will inhibit wound healing (such as in diabetic ulcers). Our treatment, like the other treatments using the topical antimicrobial creams only last 4 to 7 days, not 4 – 7 months. Overall, this should not be a problem.
Regarding the action mechanism of Fe(8-hq)3 in killing the SA, more studies need to be done, but the authors shall not make any hypothesis in discussion without basis. In addition, a hypothesize claim shall not be included in keyword, which is misleading the readers. Thus, I suggest to remove the "Fenton Reaction" from keyword.
Answer: based on the above discussions we hope this reviewer is convinced that we have conducted several solid experiments to prove the antimicrobial mechanism of Fe(8-hq)3 in killing the SA.
Similar phenomena was found in the ESI-MS data where the authors claimed that Fe(8-hq)3 is stable after incubating in cell culture medium. If stable, how to get Fe2+ to initiate the Fenton Catalytic Reaction?
Answer: we have proven that that Fe(8-hq)3 is stable in the absence of a reducing agent (similar to the extracellular environment with oxygen present) by incubating with the cell culture medium. However, in the bacterial cytosol, an enzyme called “ferric reductase” will reduce Fe3+ in Fe(8-hq)3 to Fe2+, causing the Fe2+ to be released from the complex. This enzyme is ubiquitous in bacteria as it is responsible for reducing iron from Fe3+ in all Fe-siderophore complexes into Fe2+ for releases. This is how bacteria acquire iron without enduring toxicity of iron and also makes iron bioavailable inside the cell.
In method (L413), the authors mentioned the stability test was done on bacteria culture medium, but in supplementary and results & discussions, it become cell culture medium. Please clarify.
Answer: to avoid the potential confusion, we added the word “cell” to the subtitle in the text and the word “bacterial” to the figure legend of Figure S4. See bellow:
Stability of Fe(8-hq)3 in the bacterial cell culture medium.
Figure S4. ESI/LC-MS trace of the recovered product Fe(8-hq)3 after incubating at 37 ºC and 180 rpm for 48 hours with the bacterial cell culture medium
Other than these, I am satisfied with the responses provided on my earlier comments. Thank you for the clarification and manuscript improvement.
